Calibrating abundance indices with population size estimators of red back salamanders (Plethodon cinereus) in a New England forest

Siddig Ahmed A. 1 2 asiddig@eco.umass.edu ahmedsiddig@fas.harvard.edu
Ellison Aaron M. 2
Jackson Scott 1
1 Department of Environmental Conservation, University of Massachusetts Amherst , Amherst, MA , USA
2 Harvard University, Harvard Forest , Petersham, MA , USA
Measey John
Electronic publication date: 2015 May 14
Publication date: 2015
Volume: 3
Electronic Location ID: e952
Received 2015 Feb 12; Accepted 2015 Apr 21
Copyright: © 2015 Siddig et al.
Copyright year: 2015
Copyright holder: Siddig et al.
License: This is an open access article distributed under the terms of the Creative Commons Attribution License, which permits unrestricted use, distribution, reproduction and adaptation in any medium and for any purpose provided that it is properly attributed. For attribution, the original author(s), title, publication source (PeerJ) and either DOI or URL of the article must be cited.
License URL: https://creativecommons.org/licenses/by/4.0/

Keywords: Amphibian monitoring, Indicator species, Long-term monitoring, Plethodon cinereus, Population size, Regression calibration, Removal sampling, Salamander, Tsuga canadensis, Abundance index

Funding: National Science Foundation (NSF) 0620443 1003938 1237491 Islamic Development Bank (IDB) This work is a publication of the Harvard Forest LTER and REU Sites (supported by NSF grants 0620443, 1003938, and 1237491). The senior author was supported by a scholarship from the Islamic Development Bank (IDB). The funders had no role in study design, data collection and analysis, decision to publish, or preparation of the manuscript.

==============================
Herpetologists and conservation biologists frequently use convenient and cost-effective, but less accurate, abundance indices (e.g., number of individuals collected under artificial cover boards or during natural objects surveys) in lieu of more accurate, but costly and destructive, population size estimators to detect and monitor size, state, and trends of amphibian populations. Although there are advantages and disadvantages to each approach, reliable use of abundance indices requires that they be calibrated with accurate population estimators. Such calibrations, however, are rare. The red back salamander, Plethodon cinereus, is an ecologically useful indicator species of forest dynamics, and accurate calibration of indices of salamander abundance could increase the reliability of abundance indices used in monitoring programs. We calibrated abundance indices derived from surveys of P. cinereus under artificial cover boards or natural objects with a more accurate estimator of their population size in a New England forest. Average densities/m2 and capture probabilities of P. cinereus under natural objects or cover boards in independent, replicate sites at the Harvard Forest (Petersham, Massachusetts, USA) were similar in stands dominated by Tsuga canadensis (eastern hemlock) and deciduous hardwood species (predominantly Quercus rubra [red oak] and Acer rubrum [red maple]). The abundance index based on salamanders surveyed under natural objects was significantly associated with density estimates of P. cinereus derived from depletion (removal) surveys, but underestimated true density by 50%. In contrast, the abundance index based on cover-board surveys overestimated true density by a factor of 8 and the association between the cover-board index and the density estimates was not statistically significant. We conclude that when calibrated and used appropriately, some abundance indices may provide cost-effective and reliable measures of P. cinereus abundance that could be used in conservation assessments and long-term monitoring at Harvard Forest and other northeastern USA forests.

Introduction

Amphibians are declining worldwide due to climatic changes, habitat loss and alteration, invasive species, diseases, and environmental pollution (Becker et al., 2007; Dodd, 2010); the number of threatened amphibian species increased nine-fold between 1996 and 2011 (Lanoo, 2005; IUCN, 2011). Because amphibians are physiologically sensitive to many local environmental characteristics, they are thought to be useful indicator species for monitoring local environmental changes (Welsh & Hodgson, 2013, but see Kerby et al., 2010). Thus, the overall decline of amphibians worldwide could suggest a corresponding deterioration of environmental conditions. However, indicator species can be used reliably to monitor environmental conditions and to inform conservation programs only if indices used as indicators, such as population size, reflect the actual measurement (e.g., abundance or density) of the species of interest (Yoccoz, Nichols & Boulinier, 2001).

Two standard methods are used to accurately estimate the size of amphibian populations (Heyer et al., 1994): capture-mark-recapture methods (Seber, 1982; Bailey, Simons & Pollock, 2004a; Bailey, Simons & Pollock, 2004b) and depletion (removal) methods (Zippin, 1956; Bailey, Simons & Pollock, 2004a). Although both of these methods yield reliable estimates of abundance, they are impractical to use when species have very large home ranges, low detection probability, or are cryptic or rare (Royle, 2004). Long-term monitoring programs also may not have sufficient resources to regularly (e.g., annually) repeat intensive mark-recapture or depletion studies. Finally, mark-recapture studies that rely on toe clipping or PIT tags may reduce survival and have been critiqued on ethical grounds (e.g., Clark, 1972; Heyer et al., 1994; Ott & Scott, 1999; Green, 2001; May, 2004; Dodd, 2010; Guimarães et al., 2014), and depletion studies can reduce local population sizes (Hayek, 1994).

Because of these challenges, many herpetologists and conservation biologists who use amphibians, including Plethodontid salamanders, as indicator species use indices of abundance derived from simple counts of individuals under artificial cover boards, random searching of natural objects, pitfall traps, or visual encounter surveys (Heyer et al., 1994; Mathewson, 2009; Mathewson, 2014; Welsh & Hodgson, 2013). Although abundance indices routinely are assumed to be proportional to absolute measures of abundance, assuming a constant capture probability (i.e., detectability), these indices may not provide accurate estimators of population size. For example, salamanders may be attracted to cover boards or pitfall traps, and random searching or visual encounter surveys may not provide reliable estimates of detection probability or occupancy, which also are rarely constant (e.g., Krebs, 1999; Pollock et al., 2002). Nonetheless, abundance indices often are easier to obtain than other estimators of population abundance, can be determined for large areas, are less intrusive, minimize harm to individuals, and are cost-effective (Royle, 2004; Pollock et al., 2002).

The trade-off between the need for reliable and cost-effective abundance indices versus labor-intensive but more accurate abundance estimators has led to research that combines both methods using model-based inference (e.g., Smith, 1984; Buckland, Goudie & Borchers, 2000). Two approaches are used commonly in studies of birds and mammals. N-mixture models use Poisson or binomial likelihoods of abundance indices or repeated count data to obtain site-specific estimates of abundance (e.g., Royle, 2004). Alternatively, abundance indices can be calibrated to population estimates obtained from mark-recapture or depletion studies (e.g., Eberhardt & Simmons, 1987; Brown et al., 1996). However, neither N-mixture models nor direct calibration of abundance indices have been adopted widely by herpetologists, who generally use uncalibrated abundance indices to draw inferences about population sizes and demographic rates, and then use these inferences to guide management applications (Mazerolle et al., 2007). Here, we calibrate abundance indices derived from transect surveys of counts of salamanders found under cover boards and natural objects with simultaneous estimates of local population sizes of eastern red back salamanders (Plethodon cinereus (Greene, 1818)) obtained using replicated depletion studies in a New England Forest.

This study is particularly timely because of the ongoing decline of Tsuga canadensis (L.) Carrière, a foundation tree species in New England forests (Ellison et al., 2005). Tsuga canadensis is being killed by a non-native insect, Adelges tsugae, which is spreading rapidly throughout the eastern United States (e.g., Orwig et al., 2012). Because T. canadensis has a large range, assessment of the consequences of its decline at any particular site requires rapid, fine-scale studies of the status and trends in populations of species associated with T. canadensis. For example, the loss of the majority of T. canadensis individuals from southern and central New England forests over the next several decades is expected to lead to parallel declines in salamander populations (e.g., Ellison et al., 2005; Mathewson, 2009; Mathewson, 2014). Designing, validating, and implementing a long-term monitoring program for salamanders in these forests requires both accurate base-line estimates of population sizes and methods to rapidly (re)assess populations for many years to come (e.g., Bailey, Simons & Pollock, 2004b; Mazerolle et al., 2007; Gitzen et al., 2012).

Materials and Methods

Our calibration study involved four sequential steps (Fig. 1):

1. Establishment of plots and sampling transects, and emplacement of cover boards (May 2013);

2. Simultaneous depletion sampling, surveys of natural cover objects, and surveys of cover boards (repeated twice in July 2014);

3. Estimation of population sizes from depletion sampling;

4. Regressions of data from cover board surveys and natural object surveys on estimated population size of P. cinereus.

Figure 1 Framework for calibrating salamander abundance indices with population size estimators.

Study species

Plethodon cinereus is a common woodland amphibian in the family Plethodontidae. This is the largest family of salamanders, with at least 240 species (Hairston, 1987; Mathewson, 2006; Dodd, 2010). Plethodontid salamanders, including P. cinereus, are lungless organisms that respire through their skin (Hairston, 1987). Plethodon cinereus also has no aquatic life-history stage; rather it is completely terrestrial and spends its entire 3–7 year lifetime in forested areas, living in or under moist soils, rotting logs, leaf litter rocks, and other natural cover objects. The females lay 3–14 eggs underneath moist soils and natural objects between mid-June and mid-July; the incubation period is 6–9 weeks long (Petranka, 1998). The home range of P. cinereus is relatively small (13 m2 on average), and they normally move <1 m/day when foraging for prey at the soil surface (Mathewson, 2006). Its limited mobility has suggested that P. cinereus should be an excellent indicator of changes to environmental conditions in the forested ecosystems in which they live (Welsh & Hodgson, 2013; Mathewson, 2009).

The population biology and trophic position of P. cinereus also is well studied. For example, Burton & Likens (1975) reported that the density of P. cinereus at Hubbard Brook, New Hampshire was ≈0.25 salamanders/m2, and that their total biomass was equal to that of small mammals and twice that of breeding birds at their study site. These numbers are conservative, as only 2–32% of the local population of P. cinereius normally is present on or near the surface during the warm and moist or rainy nights when this species is typically sampled (Taub, 1961; Burton & Likens, 1975). Their high abundance makes P. cinereus an important prey item of many birds and snakes, and this salamander also is a significant predator of many soil-dwelling invertebrates including insects (Welsh & Hodgson, 2013).

Study site and locations of calibration plots

This calibration study was done at the Simes Tract (Ellison et al., 2014) within the Harvard Forest Long-term Ecological Research (LTER) site in Petersham, Massachusetts, USA (42.47°–42.48°N, 72.22°–72.21°W; elevation 215–300 m a.s.l.). All measurements were taken within four separate forest stands. Two of these stands were dominated by eastern hemlock (Tsuga canadensis) and the other two were composed of mixed deciduous species, including oaks (Quercus spp.) and maples (Acer spp.) species. The two hemlock sites were in a moist valley, whereas the two deciduous locations were on a drier ridge ≈500 m from the valley. Individual stands within a forest type were separated by >100 m, so all four sites can be considered independent replicates.

Figure 2 Sampling design showing the layout of the sampling transects and arrangement of the cover boards at the Simes Tract of the Harvard Forest, Petersham, Massachusetts.

Figure 3 Cumulative numbers of salamanders captured during each depletion sampling session.

Each panel illustrates the cumulative number of salamanders captured in a single plot in either hemlock or the hardwood stands. The data for each 4-day sampling session in each plot × forest type combination are shown in different colors.

Transects for depletion sampling, natural object surveys, and cover boards were established in May 2013. Within each stand, we laid out three parallel 30 × 1-m strip transects, separated from one another by 10 m (Fig. 2). Depletion sampling and natural object surveys were done along all three transects. Along each of two of these transects (the outer ones) in each stand, we placed five cover boards (1 × 0.25 × 0.02 m rough-sawn T. canadensis planks) spaced 5 m from one another. To ensure that the lower surface of each cover board was in contact with the soil surface, leaf litter directly under the cover board was removed before the cover board was laid down. To minimize effects of the disturbance of establishing the sampling locations on detection of P. cinereus, and to allow for appropriate weathering (Mathewson, 2009; Hesed, 2012), all sampling was done in July 2014, 14 months after the sites had been selected, transects laid out, and cover boards placed in the field. Following each sampling day, all transects, including natural objects on the forest floor, were left in similar conditions to those seen at the start of the day.

Salamander sampling

Depletion sampling of P. cinereus, surveys of these salamanders under natural cover objects, and counts of individual salamanders under cover boards in all four plots occurred during two four-day sessions in July 2014. The first session ran from 14 to 17 July, and the second from 27 to 30 July. All sampling was done on the morning of each day between 0700 and 1100 h.

Depletion sampling

Our depletion sampling procedure followed that developed by Hairston (1986), Petranka & Murray (2001), and Bailey, Simons & Pollock (2004a). Every morning during each of the two four-day sampling sessions, we intensively searched for salamanders for ≈4 h under dead wood, rocks, and leaf litter in each transect in each plot. All salamanders encountered in each transect were removed and placed into 0.7 × 0.3 × 0.15-m plastic baskets buried 5 m outside of the sampling zones. The bottom 10 cm of each basket was filled with dirt and leaf litter to provide moist habitat and food; small holes were drilled in the bottom of each basket to allow rain water to drain; and baskets were covered with mesh netting to provide shade and protection from predators (Corn, 1994). All salamanders collected from the transects were kept in these baskets for the entire sampling session (up to 72 h), and were released thereafter back into the study plots from which they had been collected.

Cover-board sampling

We lifted up each cover board, counted the number of P. cinereus that we saw under it (Mathewson, 2009; Hesed, 2012), removed the salamanders from under the cover boards, and placed them in the holding baskets.

Abundance estimations and calculation of abundance indices

The three abundance estimates were calculated for each sampling session separately. From the data collected from the depletion surveys, we estimated capture probability and population size of P. cinereus in each plot using Zippin’s regression method (Zippin, 1956; Zippin, 1958) as implemented in the Removal Sampling software, version 2.2.2.22 (Seaby & Henderson, 2007). In this method, the total number of individuals captured and removed from the sampling area (i.e., each transect) each day was plotted as a function of the cumulative number of captures on previous days in the same transect. The estimated population size for each transect is defined as the point where the regression line intercepts the x-axis, and the capture probability as the slope of the regression line (Zippin, 1956; Zippin, 1958; Seaby & Henderson, 2007). Estimates of population size per m2 or per ha were obtained by division (we sampled 30 m2 per transect) or multiplication (1 ha = 10,000 m2), respectively.

A transect-level cover-board index (salamanders/m2) was estimated as the average of the number of salamanders detected during the first day of each sampling session under all five cover boards in the transect, multiplied by 4 (the area of a single cover board = 0.25 m2). Similarly, a transect-level natural object survey index (salamanders/m2; excluding the cover boards) was estimated as the total number of salamanders captured during the first day of sampling in each transect divided by 30 (the total area of strip transects searched for salamanders was 30 × 1 m2 = 30 m2). In both cases, we calculated population indices for each sampling session only from the first day of captures to avoid effects of habitat disturbance (from searching) and ongoing removal sampling on the subsequent three days of detection and capture of salamanders.

Calibration of indices

We calibrated the two density indices (from cover boards and natural objects) by regressing them against the estimates of population size derived from depletion sampling (Eberhardt, 1982).

Results

Between both sampling sessions and summed over all three sampling methods, we captured or detected a total of 101 P. cinereus individuals: 53 individuals were captured in the first sampling session and 48 in the second. There was no significant difference between the number of salamanders captured in the hemlock plots (59) and the hardwood plots (42) (Wilcoxon rank sum test: W = 24, P = 0.18). As is typically found in depletion studies, the total number of captures/day declined continuously in both forest types, and cumulative captures generally leveled off by the fourth day of sampling during each session (Fig. 3).

Figure 4 Regressions of population estimates (salamanders/m2).

The average population density of P. cinereus estimated from the depletion surveys ranged from 0.13 (hardwood) to 0.18 (hemlock) salamanders/m2 (1330 to 1816 salamanders/ha), with an overall average of 0.15 salamanders/m2 (1550/ha) (Table 1). The average capture probability in the hemlock stands was 0.51, about 15% lower than that in the hardwood stands (0.64). In contrast, the average relative density suggested by cover-board observations was 1.7 individuals/m2 in the hemlock stands and 0.7 salamanders/m2 in the hardwood stands, with an overall average of 1.2 salamanders/m2. Last, the estimated density of P. cinereus from searches of natural objects within each 30 × 1-m transects was 0.1 and 0.06 salamanders/m2 in the hemlock and hardwood stands, respectively, with an overall average of 0.08 salamanders/m2. Overall, there were no significant differences between forest stand types in any of these estimates (Table 1).

Table 1 Mean estimates (standard error of the mean) of P. cinereus population size and abundance indices.

	Forest type			
Salamanders/m2	Hemlock	Hardwood	Wilcoxon’s W	P value	
Depletion sampling	0.18 (0.03)	0.13 (0.02)	6.5	0.461	
Cover-board index	1.7 (0.4)	0.7 (0.17)	0	0.125	
Natural-object survey index	0.1 (0.02)	0.06 (0.01)	7	0.562	

Because we found no differences between forest-stand types in salamander density or abundance indices, we pooled the data from the two forest-stand types when we calibrated the two indices using the estimated population density (Fig. 4). The estimated true density of P. cinereus was predicted well by the natural-objects survey (r2 = 0.65, P = 0.001; Fig. 4) but the cover-board index was weakly and not significantly associated with the estimated true population density (r2 = 0.30, P = 0.158). The density index from the natural object survey underestimated the estimated population density of P. cinereus by 50%, whereas the cover-board index overestimated the estimated population density of P. cinereus by a factor of eight (Fig. 4).

Discussion

Estimation of the abundance of organisms is at the core of population biology and conservation practice (Krebs, 1999). However, in spite of the importance of accurate estimates of population size, many ecologists and environmental scientists use abundance indices that rarely are calibrated with actual abundance data. We have shown here that, with only modest effort, at least one abundance index for P. cinereus can be calibrated reasonably well, allowing for stronger inferences regarding salamander population size.

Our results represent the first time, to our knowledge, that an abundance index of salamander population size has been calibrated to actual density estimates in northeastern North America. Our results suggest that rapid surveys of natural cover objects in two forest types (hemlock or mixed deciduous stands) correspond reasonably well with estimates of population size obtained from more careful, labor-intensive depletion samples. Our results also were similar to relative abundance of P. cinereus found during cover-board surveys a decade ago at Harvard Forest (Mathewson, 2009). However, our estimates of abundance from depletion sampling (1,816 salamanders/ha) were 20% lower than those found in hardwood forests at Hubbard Brook, New Hampshire (2,243 salamanders/ha; Burton & Likens, 1975). Both of these density estimates are likely to be quite conservative, as Taub (1961) suggested that only 2–32% of a local population of P. cinereus is available for sampling on the soil surface or within the topsoil during a given period of time.

Although the abundance index obtained by natural object surveys was well calibrated with the population size estimator from depletion sampling, the cover-board index was not well calibrated. The overestimation of population density suggested by cover board surveys were not surprising, as cover boards provide additional protected habitat at the soil surface that should be attractive to P. cinereus (Hesed, 2012). The spatial heterogeneity in P. cinereus individuals and their relatively low mobility also may have contributed to the large variability in the cover-board index (CV = 77%; Table 1). Overall, we conclude that population indices of P. cinereus from natural objects surveys are more reliable than indices from cover-board surveys within our study area.

Calibrating indices with population density estimation using methods such as removal sampling requires that all the different sampling methods be done simultaneously over a large area, a process that is labor (and hence, cost) intensive. If salamander sampling is part of a long-term monitoring program, we recommend that calibration should occur regularly. If consistent results are achieved with a series of annual calibrations, it is possible that, longer times between re-calibrations, perhaps every 4–5 year could be considered to capture the effects of, for example, changing environments. We also note that we used linear relationships to calibrate population indices with density estimates but the relationship between density and abundance indices may be non-linear (Pollock et al., 2002).

In summary, our results suggest that once they are calibrated, meaningful data on amphibian abundance may be obtained from natural object surveys that take fewer supplies, people, and time than repeating more intensive, invasive, or destructive methods (e.g., capture-mark-recapture surveys, pitfall traps, or depletion surveys). Although our data and calibrations are applicable only to the forest we studied in central Massachusetts and its particular weather conditions, the method for calibrating abundance indices is generalizable to any site. We recommend that any abundance index be routinely recalibrated just as one would do with an electronic sensor. Such calibrated abundance indices could lead to cost-effective indicators that are straightforward to implement in large-scale conservation programs and broader ecological research (e.g., Noss, 1990; Gitzen et al., 2012, or the U.S. Geological Survey’s Amphibian Research and Monitoring Initiative: http://armi.usgs.gov).

We thank Allyson Degrassi (University of Vermont) and the six undergraduate researchers who participated in this project during the 2014 Harvard Forest Summer Research Program in Ecology–Alison Ochs, Claudia Villar-Lehman, Simone Johnson, Ariel Reis, Jessica Robinson, and Joel van de Sande—for helping us with intensive field work and data collection. Two anonymous reviewers and the academic editor at PeerJ provided useful comments on an earlier version of the manuscript. All field sampling protocols were approved by Harvard University’s Institutional Animal Care and Use Committee, File 13-02-144 - June 02, 2014.

Additional Information and Declarations

Competing Interests

Author Contributions

Animal Ethics

Data Deposition

Aaron M. Ellison is an Academic Editor for PeerJ.

Ahmed A. Siddig conceived and designed the experiments, performed the experiments, analyzed the data, wrote the paper, prepared figures and/or tables, reviewed drafts of the paper.

Aaron M. Ellison conceived and designed the experiments, analyzed the data, contributed reagents/materials/analysis tools, wrote the paper, prepared figures and/or tables, reviewed drafts of the paper.

Scott Jackson conceived and designed the experiments, analyzed the data, wrote the paper, reviewed drafts of the paper.

The following information was supplied relating to ethical approvals (i.e., approving body and any reference numbers):

All field sampling protocols were approved by Harvard University’s Institutional Animal Care and Use Committee, File 13-02-144 - June 02, 2014.

The following information was supplied regarding the deposition of related data:

Harvard Forest data archive: http://harvardforest.fas.harvard.edu:8080/exist/xquery/data.xq?id=hf246

HF Data Archive ID = HF246.

DOI = 10.6073/pasta/9a1f20f06e6674aade200fcadf42f66e.

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
