# Peer review of "Calibrating abundance indices with population size estimators of red back salamanders (Plethodon cinereus) in a New England forest"

_PeerJ, doi:10.7717/peerj.952_

## Round 0.1 · original submission · Major Revisions

The manuscript contains interesting and valuable information concerning the validation of different methods. There are a number of points raised by the reviewers concerning both the clarity of the methodological description (which is essential in a paper such as this) and about the interpretation of data (including/excluding outliers). In addition, I agree that there are some generalisations made in the text that are not entirely supported by the literature cited, and I'd encourage the authors to go back to the original sources and check statements carefully. It would also be useful to have some discussion of how widely the results herein could be used. Clearly, the authors have in mind to continue this work to monitor the spread of an invasive species, but an honest critique of whether this approach would be valid at other sites would be useful.

Reviewer 1 ·

Basic reporting

This study is focused, well motivated and formatted correctly. To the best of my knowledge this article meets the basic reporting standards of PeerJ.

Experimental design

The experimental design and analyses are generally clear and appropriate. One aspect that might benefit from further elaboration is the degree of independence of the plots within forest types. My first reading made me think that the 4 replicates were all within a single stand, and this made me wonder if they had pseudo-replicated the forest type comparison. I think from looking at the map that the plots probably are good replicates and not overly spatially confounded - but I would appreciate further elaboration on this point.

Validity of the findings

The main weakness here is the cover board outlier. With an n of only 8 and its position of strong leverage make it such that this one point has a huge impact on the results. Treat it "as is" and coverboard sample is very weakly related to density estimates, omit or use median and there is a good relationship. For cover boards it all hinges on that one point, unfortunately. I think they handled it as well as they can, given the data. However, they have a strong relationship with the natural cover surveys and it makes sense to me that the natural cover and cover board relationships to density estimate would be similar. So I still think there are worthwhile robust results, even if the cover board result is depended on how they handle the outlier.

Additional comments

I found this well written and straightforward. Handling of the literature was, I thought, very scholarly.

Reviewer 2 ·

Basic reporting

Additional editing is needed. The title and abstract feature several mistakes in capitalization, subject-verb agreement, punctuation, and usage; errors continue at a lower frequency through the body of the manuscript. Inconsistent capitalization is particularly an issue with common names of species referred to in the text. If the authors are too familiar with the text to catch these errors, a friend or colleague or campus writing center should easily be able to help.
20: Artificial coverboards (inconsistently called “cover boards” or “cover-boards” in the manuscript) are not an abundance index. Abundance indices are counts of salamanders.
22–23: What do advantages and disadvantages of the approaches have to do with calibration? Why “despite”?
25, 31, 32: capitalization
26, 88: The salamander’s scientific name is incorrectly attributed to Linnaeus.
26–28: Subject-verb agreement
29: Unclear subjects: “density of searches and observations”
47: number of amphibians or amphibian species?
48–51: Disagreement in the literature about this claim should be discussed; this is not a universally accepted statement, and the implications of the manuscript relate directly to this claim (e.g., Kerby, J. L., K. L. Richards-Hrdlicka, et al. (2010). "An examination of amphibian sensitivity to environmental contaminants: are amphibians poor canaries?" Ecology Letters 13(1): 60–67.).
54–56: The claim that these methods are commonly used is not supported by any examples.
60–62: Multiple studies of skin-marking (VIE) indicate no negative effects; I don’t know of any negative effect that has been shown. Concern about toe-clipping (a marking method) does not invalidate mark-recapture studies (a monitoring approach).
62–63. The citation relates to commercial trade, not scientific studies. These are substantially different. Although removal studies are a central focus of this manuscript, there is no relevant literature review.
65: capitalization, usage; these problems continue in the manuscript, but I won’t keep pointing them out.
64–67: “many herpetologists and conservation biologists who use amphibians as indicator species use indices of abundance”: the only citations are a 20-year old reference manual and a technical description of a population size estimator. These claims need support from a review of the pertinent literature. It doesn’t have to be an exhaustive review, but these are statements that would get a “citation-needed” flag on Wikipedia—contributions to the scientific literature need to be held to a higher standard.
The text in Figure 1 is too small and low-resolution to be legible; same for the axes on Figure 3.

Experimental design

118–126: When the boards were placed relative to sampling and how they were placed with respect to the ground are not described; both are important (content of Miller Hesed 2012, cited later in the manuscript, should be considered here).
129: The Bailey et al. 2004 paper cited as a reference for the depletion procedure has nothing to do with removals. I really question the good-faith effort of literature review that went into this manuscript. This concern is not about impact; it has to do with surveying and understanding what work has previously been done in order to inform the design of the study and the interpretation of its results. The content of the manuscript does not indicate that this has been done, although claims are made and references are cited as though it has been.
Marsh and Goicochea (Marsh, D. M. and M. A. Goicochea (2003). "Monitoring terrestrial salamanders: biases caused by intense sampling and choice of cover objects." Journal of Herpetology 37(3): 460–466.) discussed effects of sampling plots more often than once per week. The study design in this manuscript likely had a strong effect on the numbers of salamanders encountered under the coverboards.
Before describing the depletion sampling and the coverboard sampling separately, clarity would be greatly improved by a few sentences giving an overview of the process. I had to read 2.1 and 2.2 repeatedly to try to understand what was done, and even then it wasn’t clear. A clear summary of the survey/sampling methodology with justification (why were things done this way?) is badly needed.
Treating coverboards as natural objects on two of the transects confounds the comparison they are attempting to make.
The timing of the sampling procedure is not clearly described.
153: Capture probability has meaning in the extensive mark-recapture literature, and this is not it. Actual detection probability does not seem to have been estimated in this study, and it should be expected to have a large impact on the results.

Validity of the findings

Based on their sampling design and the biology of Red-backed Salamanders (not discussed at all), their depletions over four consecutive days likely indicated almost nothing about actual population size; zero captures on the fourth day almost certainly means that their survey methods decreased the detection probability of salamanders, not that they removed all of the salamanders from the plot.
The discussion introduces new results (salamanders / ha) not previously reported in the manuscript; there is no description of how they were calculated or how they should be interpreted based on the study design.

Additional comments

I am not convinced that the results of this study would be applicable to any other site, and I am not sure that they say anything meaningful about the study plots themselves. The study is extremely limited spatially and temporally, and sample sizes on each plot are also extremely small. The intensive sampling likely affected salamander behavior, and not accounting for detection probability limits your ability to say anything meaningful about true salamander abundance, which is central to the manuscript as it is currently framed. There have been many well-designed studies comparing different survey methods for salamanders. You should read these and carefully consider whether this study can add to that literature. You may be able to reframe this manuscript as a simple comparison between forest types, once all of the other issues are addressed.

---

## Round 0.2 · accepted · Accept

Although one reviewer calls for continued changes to your ms to acknowledge limitations in sampling, I feel that this is currently covered in the existing ms.

Reviewer 1 ·

Basic reporting

I felt the manuscript was clearly written. I felt background was sufficient and the objectives worthy of attention and well motivated.

Experimental design

The research question was relevant and meaningful. The general framework of the approach was solid in many respects but suffered the following weaknesses:

-limited replication
-limited temporal scale (study was largely done over 8 days)
-limited spatial scale - replication of 2 at forest type scale
-they have no estimate of detection - so even depletion only provides a index of abundance, it could be argued

The research conforms to ethics standards for vertebrate research

Validity of the findings

The point of the paper is that abundance indices are low cost and low impact compared to more intensive methods but suffer in that they are not easily linked to more quantitative abundance measures. They authors pose that by calibrating indices we may find a happy medium, maintaining their benefits and making them more robust. This is a good motivation I feel.

The weaknesses of the paper are that it is not that clear whether the depletion sampling gives a good estimate of the "true" population size due to limited temporal scope and not accounting of detectability and also that the field work was of limited spatial scope and so comparisons between forest types have low power. Finally the limited replication means that some results (notably cover boards) are very sensitive to outliers.


However, the authors acknowledge these weaknesses and address them as best they can - short of collecting more data, For PeerJ decisions are not made based on any subjective determination of impact, degree of advance, novelty, being of interest to only a niche audience, etc. There is nothing "wrong" per se with what they present, in my opinion.

Reviewer 2 ·

Basic reporting

Relevant prior literature is still lacking. There is a large literature addressing variability in sampling techniques for monitoring salamander populations; the results of these papers are central to the claims of the manuscript, but they are not addressed. Mazerolle et al. 2007, although cited, does not seem to have been read.

Experimental design

The major problem with this manuscript is that it is making claims about accuracy that cannot be addressed by the methods. The authors use two methods for counting salamanders in a population of unknown size: removing them or not removing them. They then assume that one of those methods, removals, provides accurate estimates of population size. They have no evidence to support this claim, which is the central claim of the manuscript, nor could they possibly address it given the methods of the study. In order to compare the accuracy of sampling methods (including removals), they would need to start with a population of known size. This could very easily be done in a mesocosm setup, and it was done more than 50 years ago by Taub. Several studies have shown that the proportion of terrestrial salamander populations on the surface at any given time is variable and generally small. The connection between counts of salamanders and detection probabilities does not seem to be understood in the manuscript: removing X salamanders from under cover objects in a few days of sampling does not mean that there were X salamanders present. Due to low detection probabilities and the effects of intensive sampling (addressed in the manuscript but apparently not understood), the actual number of salamanders present is almost certainly much larger than the number removed. The authors are using a method that underestimates population size (removals), then assuming that this is the accurate population size, and then concluding that methods which generate a larger estimated population size are overestimates. Although this approach is almost certainly inaccurate, they have no data with which to even address the accuracy or inaccuracy of their methods.

Validity of the findings

Addressed above.

Additional comments

Again: you should really look into the literature comparing methods of sampling salamander populations, and seriously consider what your study design can add to it. You have collected data which could provide some interesting information, but they cannot support the claims you are using them to make.